# Soil warming during winter period enhanced soil N and P availability and leaching in alpine grasslands: A transplant study

**Eva Kaštovská[1]***, **Michal Choma[1]**, **Petr Čapek[1]**, **Jiří Kaňa[1,2]**, **Karolina Tahovská[1]**, **Jiří Kopáček[2]**

1 Faculty of Science, Department of Ecosystem Biology, University of South Bohemia in České Budějovice, České Budějovice, Czech Republic, 2 Biology Centre of the Czech Academy of Sciences, v.v.i., Institute of Hydrobiology, České Budějovice, Czech Republic

* ekastovska@prf.jcu.cz

## Abstract

Alpine meadows are strongly affected by climate change. Increasing air temperature prolongs the growing season and together with changing precipitation patterns alters soil temperature during winter. To estimate the effect of climate change on soil nutrient cycling, we conducted a field experiment. We transferred undisturbed plant-soil mesocosms from two wind-exposed alpine meadows at ~2100 m a.s.l. to more sheltered plots, situated ~300–400 m lower in the same valleys. The annual mean air temperature was 2°C higher at the lower plots and soils that were normally frozen at the original plots throughout winters were warmed to ~0°C due to the insulation provided by continuous snow cover. After two years of exposure, we analyzed the nutrient content in plants, and changes in soil bacterial community, decomposition, mineralization, and nutrient availability. Leaching of N and P from the soils was continuously measured using ion-exchange resin traps. Warming of soils to ~0°C during the winter allowed the microorganisms to remain active, their metabolic processes were not restricted by soil freezing. This change accelerated nutrient cycling, as evidenced by increased soil N and P availability, their higher levels in plants, and elevated leaching. In addition, root exudation and preferential enzymatic mining of P over C increased. However, any significant changes in microbial biomass, bacterial community composition, decomposition rates, and mineralization during the growing season were not observed, suggesting considerable structural and functional resilience of the microbial community. In summary, our data suggest that changes in soil temperature and snow cover duration during winter periods are critical for altering microbially-mediated processes (even at unchanged soil microbial community and biomass) and may enhance nutrient availability in alpine meadows. Consequently, ongoing climate change, which leads to soil warming and decreasing snow insulation, has a potential to significantly alter nutrient cycling in alpine and subalpine meadows compared to the current situation and increase the year-on-year variability in nutrient availability and leaching.

**Data Availability Statement:** All relevant raw data about soil chemistry and biochemistry are within the paper and its Supporting Information files. The raw sequencing reads were deposited in NCBI

Sequence Read Archive (SRA) database under
BioProject PRJNA800209 as submission
SUB10992790.

**Funding:** The study was financed by the Czech
Science Foundation (project no. 20-19284S)
awarded to JK, URL: https://gacr.cz/en/. The funder
had no role in the study design, data collection and
analysis, decision to publish, and preparation of the
manuscript.

**Competing interests:** The authors have declared
that no competing interests exist.

# Introduction

The ongoing climate change puts pressure on the functioning of current ecosystems. Signs of global climate change can already be observed by the widespread melting of the alpine glaciers [1,2] and the migration of plant and animal species towards the poles or higher elevations [3,4]. These changes are attributed to the combined effects of increased global temperatures, altered precipitation patterns, and consequent changes in the soil moisture regime, presence and duration of snowpack [5–8], and prolonged vegetation season [9]. The warming trend is expected to continue: the annual average air temperature is expected to increase by 2.5–4°C by the end of this century [10], with more pronounced changes at higher elevations and latitudes [11,12].

European mountain grasslands in the alpine and sub-alpine zones are already highly impacted by ongoing climate change [13,14]. This is shown by significant changes in their plant communities [15], commonly connected with increases in plant productivity [16,17] and tissue nutrient concentrations [18]. The observed vegetation changes are necessarily associated with altered processes supplying plants with nutrients. The microbially mediated soil processes involved in carbon and nutrient cycling are impacted both directly by the temperature and moisture changes [16,18,19] and indirectly through the changing plant productivity and species composition [20]. In concert with climate change-mediated vegetation changes, many studies have shown accelerated N mineralization [16,21–24] and elevated availability of inorganic N [21] in arctic and alpine soils. These changes in soil nutrient cycling could be accompanied by adaptations of soil microbial biomass [25] and its community structure [26] to changing conditions. On the other hand, the long-term resistance of microbial communities and enzymatic activities to climate change is also often documented [19,25,27]. Generally, the responses of belowground compartment of ecosystems to climate changes are more complex and variable compared to the reaction of vegetation [28].

A typical characteristic of the seasonally snow-covered alpine and sub-alpine grasslands is their strong seasonality in vegetation activity [29] and structural and functional composition of microbial communities mediating nutrient transformation in the soil [30–32]. The functioning of alpine meadows during the vegetation season is driven by plant photosynthetic activity, connected with the release of organic compounds by roots to the soil, intensive microbial decomposition and mineralization processes and fast plant nutrient uptake and immobilization. During the winter period, activities in the belowground ecosystem compartment are largely controlled by the presence, depth and continuity of snow cover, which insulates the soil and regulates its temperature [33]. A deep continuous snowpack prevents subzero temperatures [34], allowing enzymatic decomposition, N mineralization, and nitrification to keep occurring at significant rates. The N released over winter is both immobilized in the microbial biomass [35,36] and accumulated in mineral forms in the soil [37,38]. In contrast, a shallow or discontinuous snowpack may enable soil freezing, which significantly reduces, although does not completely stop, microbial activity [37]. However, more frequent freeze-thaw events may lead to microbial cell lysis and plant root damage and mortality, associated with high rates of soil N mineralization and nitrate leaching during these events [39–41]. A spring snowmelt is a dynamic period in alpine systems. The winter microbial biomass collapses [42], the community switches to species utilizing fresh plant-derived inputs [36,42], and plants regrow. Percolating water from melting snow mobilizes and leaches available forms of C and nutrients, especially accumulated nitrate from the soils [38,41,43].

Nitrogen retention and losses in alpine meadows are controlled by a balance between N mineralization and nitrification/denitrification during the winter season, snowmelt timing, amount of percolating water and plant nutrient uptake on the onset of the growing season

[8,44,45]. Particularly, the winter and spring events, which influence the presence, depth and duration of the snow cover, strongly impact nutrient cycling, as well as the vegetation of alpine and arctic ecosystems [6]. Warming during the vegetation period further modifies the functioning of ecosystems through altering plant productivity [46], vegetation composition [47], and associated soil microbial processes. Most observations to date show a change in the precipitation pattern, reduction in snowfall, thinning snow cover, greater temperature fluctuations in winter, earlier snowmelt and a prolonged vegetation season in alpine and arctic ecosystems [8,48]. However, the ongoing climatic changes, as well as the observed responses are not uniform but rather site-specific [49]. Therefore, *in situ* transplant studies [50] or field manipulation studies [36,40,41] on regional scales, which cover different summer and winter climate change scenarios are needed [49]. These integrate multiple factors with their annual dynamics altered by climate change, which are difficult to simulate in the laboratory [28,51] and thus impede the assessment of their impacts on natural ecosystems.

The aim of our study was to investigate the potential effects of climate change on soil nutrient cycling and leaching from alpine meadows. We located our experiment in the Tatra Mountains (Slovakia), which exhibited pronounced changes in climate characteristics from 1996 till 2019, including an increase in mean annual air temperature from 2.2 to 3.5°C, prolonged growing season, increasing frequency of winter thaws, increased annual precipitation and intensity of rainfall events [52,53]. The climate changes accelerated erosion and weathering in rocky areas, which elevated leaching of base cations, bicarbonate, and phosphate into alpine lakes [53] and supported their biological recovery from previous atmospheric acidification [54]. To simulate climate change, we used the environmental gradients along the slopes of the Tatra Mountains and transferred the undisturbed plant-soil cores (hereafter referred to as mesocosms) from high- to low-elevation and warmer sites for two years. We hypothesized that: (1) The simulated "warming" will enhance rates of enzymatic decomposition and increase mineralization of nutrients, their availability in the plant-soil mesocosms, and leaching; and (2) the climate-related change in soil nutrient cycling will be accompanied by an adaptation of the soil microbial biomass and shift in bacterial community composition. Besides relatively well described climate change-related impacts on N cycling, we also focused on the rarely studied [e.g., 55] P availability and leaching from plant-soil systems.

## Material and methods

### Study area and characteristics of the transplant sites

The study was conducted in the Tatra Mountains situated in central Europe at 49.2° N, 20.2° E. The two-year transplant experiment simulating the effect of climate change on alpine meadow ecosystems was done on the slopes of two alpine valleys Furkotska (FU) and Velka Studena (VS), located in the southern slope of the central Tatra Mountains (Slovakia). Both valleys have granodiorite bedrock with acidic entisols (dominated by shallow, undeveloped leptosols and regosols, $pH_{H_2O}$ <4.5) of relatively low effective cation exchange capacity, ranging between 96–157 meq kg$^{-1}$ (one equivalent is one mole of charge) and low base saturation $\leq$ 25% (Table 1), covered by dry alpine meadows. The meadow area/patches are surrounded by large scree areas in the steep slopes [56].

In both valleys, we selected vegetated sites located at ~2100 m a.s.l. and below 1800 m a.s.l. While the high-elevation (H) sites represented typical alpine meadows with discontinuous short-stemmed vegetation among rocks and stones, dominated by *Luzula alpino-pilosa*, *Agrostis rupestris*, and *Juncus trifidus*, the low-elevation (L) sites were subalpine meadows with more-or-less continuous vegetation occurring among dwarf pine patches. The FU-L site had short-stemmed vegetation with dominant *Nardus stricta* and was more exposed to sunshine

**Table 1. Basic characteristics of the studied sites at high (H) and low (L) elevations in the Furkotska (FU) and Velka Studena (VS) valleys.**

| Characteristic | Units | Furkotska valley (FU) | | Velka Studena valley (VS) | |
|---|---|---|---|---|---|
| | | High (FU-H) | Low (FU-L) | High (VS-H) | Low (VS-L) |
| Elevation | m a.s.l. | 2140 | 1730 | 2100 | 1800 |
| Slope | % | 15 | 30 | 18 | 25 |
| Exposition | Degree | 230 | 165 | 125 | 195 |
| MAST (min, max) | ˚C | 1.6 (-14.5, 20.2) | 5.4 (-0.3, 22.6) | 3.7 (-14.7, 25.6) | 4.1 (-6.5, 19.8) |
| MST (veg/ nonveg) | ˚C | 5.5/-2.8 | 9.4/0.8 | 8.4/-1.3 | 7.6/0.3 |
| $pH_{H2O}$ | | 4.21 | 4.14 | 4.25 | 4.25 |
| C | mg g$^{-1}$ | 130 | 149 | 69 | 130 |
| N | mg g$^{-1}$ | 8.80 | 9.47 | 4.38 | 9.88 |
| P | mg g$^{-1}$ | 0.97 | 1.04 | 0.47 | 1.12 |
| CEC | meq kg$^{-1}$ | 104 | 146 | 144 | 140 |
| BS | % | 17 | 9 | 6 | 25 |

Abbreviations: MAST, mean annual soil temperature with absolute minimum and maximum values in brackets; MST, mean soil temperature in vegetation (May–October) / winter (November–April) season. $pH_{H2O}$, soil pH in water extracts; C, N, and P, total concentrations of carbon, nitrogen and phosphorus in soil; CEC, effective cation exchange capacity; and BS, soil base saturation.

than the shaded and wetter VS-L site dominated by *Callamagrostis villosa*. The site characteristics are listed in Table 1.

The study was created as part of the research "Deglaciation and postglacial climate development of the High Tatra Mountains" permitted by Ministry of Environment of the Slovak Republic, District Office Prešov, Permission No.: OU-OSZP1-2016/033834-006/SJ, valid for years 2017–2020.

## Design of the transplant experiment

In September 2013, six undisturbed soil cores (16 × 16 cm), 10 cm deep and with undisturbed native vegetation were excavated using a spade in the H sites and placed in plastic boxes. The bottoms of the boxes were perforated by evenly spaced circular holes (5 mm diameter), covered by a polyamide net with a mesh size of 0.5 mm, which enabled leaching of soil solution from the mesocosm. Four ion-exchange resin traps (parallels with two different ion-exchange resins) were placed beneath the bottom of each box, covering the entire bottom (Fig 1A). We used these types of resins: a mixed-bed anion-cation resin (Purolite C100E and Purolite A520E, mixed 1:1) designed to retain nitrate and ammonium N ($NO_3$-N, and $NH_4$-N) [57], and a hybrid anion resin (Purolite FerrIX A33E) for trapping phosphate ($PO_4$-P) [58] leached from the mesocosm. Then, the box was placed into a second plastic box of the same size, but without a bottom, which enabled natural seepage of water from the mesocosm to lower soil horizons. Moreover, walls of the lower box provided a physical barrier, protecting resins from the accumulation of ions from the mesocosms' surroundings. The complete mesocosms were returned to the soil (Fig 1B). At each sampling site, three mesocosms were placed to the same location serving as controls (H–high site). Three others were transferred down-the-slope to L sites (H→L–moving from high to low location) in the respective valley. Mesocosms were kept in field for two years until September 2015, when they were resampled, placed in plastic bags and kept at 4˚C for 2 days during the transport to the laboratory before analyses (see below).

Soil temperature was measured near the mesocosms at a depth of 2 cm, using HOBO® UTBI-001 sensors (Onset Computers, USA), with an accuracy of ±0.2˚C. Prior to use, the thermometers were kept several days at 4˚C and then at a laboratory temperature of ~20˚C, and

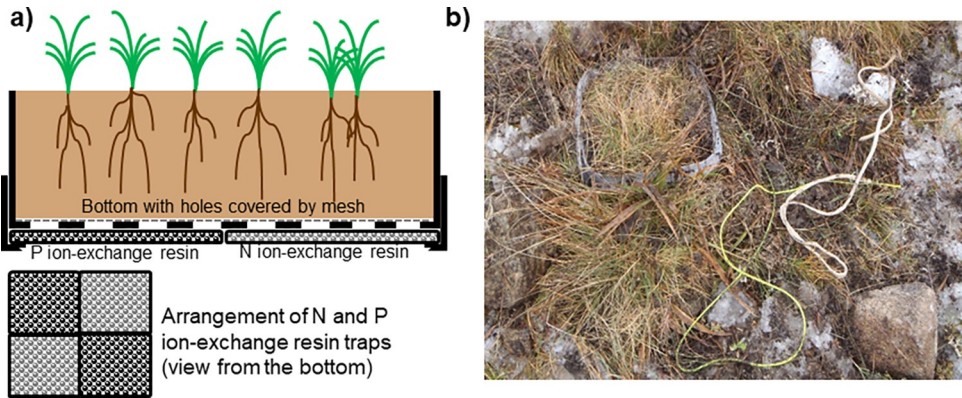

**Fig 1. Design of mesocosms.** (a) The undisturbed plant-soil core was placed in the plastic box with perforated bottom. Four ion-exchange resin traps–parallels with two different ion-exchange resins that trapped N and P, were placed beneath the core, covering the entire bottom, and secured by another plastic box of the same size but without bottom. (b) Photo of mesocosm in the field together with a thermometer in its vicinity, marked with colored string.

only thermometers with the proven accuracy range of ±0.2°C at both temperatures were used. The temperatures were recorded in 60-min intervals and were used to calculate daily mean soil temperature.

### *In situ* measurement of mineral N and P leaching from mesocosms by ion-exchange resin traps

Ion-exchange resin traps were kept in field for one year and then replaced. Their field exposure thus lasted from Sept 2013 to Sept 2014 and from Sept 2014 to Sept 2015. In laboratory, the resins removed from the traps were washed in demineralized water and extracted by repeated elution procedure in a glass column. A 10% NaCl solution was used as the elution solution for the mixed-bed resin retaining mineral N forms [57], while 2% NaOH was used for the resin retaining phosphate [58]. The final volume of eluate was 400 and 300 ml per column for mineral N and P, respectively. The concentrations of $NO_3$-N and $NH_4$-N were determined using a flow injection analyzer (FIA, Lachat QC8500, Lachat Instruments, USA), while $PO_4$ was measured as soluble reactive P (SRP) [59]. The amounts of leached nutrients were corrected for elution efficiency [57,58]. The amounts of N and P, respectively, from the two parallel ion-exchange resin traps located under the particular mesocosm were always tallied up and represented the amount of N and P leached from the half the area of the mesocosm per year. The N and P leaching was finally expressed in mg m$^{-2}$ year$^{-1}$.

### Analyses of mesocosms and ambient soils at the end of experiment in 2015

After removal in September 2015, mesocosms were analyzed for nutrient concentrations in the aboveground and belowground plant biomass, soil chemical characteristics, microbial biomass, enzymatic activities, C and N mineralization rates and bacterial community composition (see below). The vegetation and soil samples were prepared as follows. Plants were manually separated from the soil and washed in water. All the soil from the mesocosm was homogenized, sieved through a 5-mm mesh and subsequently analyzed as described below.

At the same time, the ambient soils from the original H and L localities were sampled. Three representative soil samples, each combined from five ~50–70 g soil subsamples, were taken by a small spade to the soil depth of 10 cm randomly from the alpine meadows close to the mesocosms. These samples were processed identically to the mesocosm soil, except for the bacterial community composition.

The characteristics of ambient soils from H sites were compared to those of the mesocosms' soil samples placed back in H sites to check for the "placing-to-box" effect. A comparison of vegetation and soil characteristics from the *in situ* H and downward transferred H→L mesocosms were used to assess the effects of simulated climate change.

All soil chemical and biochemical results further reported in this study are given on a dry weight basis (105°C).

**Plant biomass C, N and P contents.** All the above- and belowground plant biomass was dried at 60°C for 72 hours and milled. Total C and N concentrations were determined by dry combustion on an elemental analyzer (ThermoQuest, Italy). Total P was measured colorimetrically by the ammonium molybdate-ascorbic acid method on a flow injection analyzer (FIA, Lachat QC8500, Lachat Instruments, USA) after perchloric acid digestion [60].

**Analyses of ambient and mesocosm soils.** The soils were divided immediately after sieving for further analyses. A soil subsample (5 g) was immediately frozen for DNA extraction and an assessment of potential enzymatic activities. A part of the soil was dried at 60°C to constant weight, milled and analyzed for total concentrations of C, N and P contents using the same methods as for plant material.

Exchangeable base cations (BC = sum of $Ca^{2+}$, $Mg^{2+}$, $Na^+$, $K^+$) and exchangeable acidity (the sum of exchangeable $Al^{3+}$ and $H^+$) were determined at natural soil pH by extracting 2.5 g of air-dried soil with 50 ml of 1M $NH_4Cl$ and 1M KCl, respectively, in three successive steps [56]. Base cation concentrations were measured by atomic absorption spectrometry (Varian, Australia), and $Al^{3+}$ and $H^+$ were determined by titration (phenolphthalein, 0.1M NaOH and 0.1M HCl) [61]. The effective cation exchange capacity (CEC) was the sum of BC, $Al^{3+}$ and $H^+$. Base saturation (BS) was calculated as the percentage of BC in CEC.

Subsamples of fresh soils (10 g, 3 replicates) were immediately extracted by 40 ml of cold demineralized water on a roll-and-roll shaker for 1 hour at 4°C. The extracts were centrifuged at 4000 g for 30 min. The $pH_{H2O}$ was measured in the supernatant using a glass electrode. The supernatant was then filtered through 0.45 μm glass-fiber filter and analyzed for dissolved organic C (DOC) and total dissolved N (DN) using a TOC-L analyzer equipped with the total N measuring unit TNM-L (Shimadzu, Tokyo, Japan), and for SRP, $NO_3$-N and $NH_4$-N using flow injection analyzer (FIA Lachat QC8500, Lachat Instruments, USA).

Another set of soil samples (5 g, 6 replicates) was weighed in sealed, airtight 100 ml flasks and incubated at 20°C. In three selected samples, soil $CO_2$ efflux was measured after 2, 7, 16, and 24 days of incubation using the Agilent 6850 GC system (Agilent Technologies, CA, USA). Flasks were ventilated by a stream of air using a fan after each measurement and closed again. Cumulative respiratory C loss was calculated to characterize the potential C mineralization. The soil samples were further used to assess the net rates of potential N mineralization and nitrification. These were calculated from differences in concentrations of $NH_4$-N and $NO_3$-N, respectively, in the 0.5M $K_2SO_4$ soil extracts between the 7th and 14th day of incubation at 20°C (3 replicates extracted each time, measured using flow injection analyzer; FIA Lachat QC8500, Lachat Instruments, USA).

Microbial biomass carbon (MB-C), nitrogen (MB-N), and phosphorus (MB-P) were determined by the chloroform fumigation-extraction method [62–64] in fresh samples within 48 hours after sieving. Samples were extracted either by 0.5M $K_2SO_4$ (1:4 w:v; MB-C and MB-N) or 0.5M $NaHCO_3$ with a pH of 8.5 (1:15 w:v; MB-P) before and after a 24 hour fumigation with ethanol-free chloroform. The DOC and DN concentrations in the soil extracts were measured using TOC-L analyzer equipped with the total N measuring unit TNM-L (Shimadzu, Tokyo, Japan). The SRP in sodium bicarbonate extract was measured colorimetrically by the ammonium molybdate ascorbic acid method [59]. The MB-C, MB-N, and MB-P concentrations were calculated as differences between the respective C, N, and P concentrations in

extracts from the fumigated and non-fumigated samples and corrected for incomplete recovery, applying correction factors of 0.45, 0.54 and 0.4, respectively [62–64].

Potential extracellular enzyme activities were determined by microplate fluorometric assays. For determination of hydrolytic enzyme activities, 0.5 g of thawed soil was suspended in 50 ml of distilled water and sonicated for 4 minutes to disrupt soil particles. Then, 200 μl of soil suspension was added to 50 μl of methylumbelliferyl substrate solution specific to β-glucosidase (BG), cellobiosidase (CB), phosphatase (AP), or N-acetylglucoseaminidase (NAG) determination. To determine leucine aminopeptidase (LAP), 200 μl of soil suspension was added to 50 μl of 7-aminomethyl-4-coumarin substrate solution [65]. Plates were incubated at 20˚C for 2 hours. Fluorescence was quantified at an excitation wavelength of 365 nm and emission wavelength of 450 nm, using the INFINITE F200 microplate reader (TECAN, Germany). The method was optimized for soil samples according to German et al. [66]. All enzymatic activities were tallied and proportions of C acquisition (BG and CB), N acquisition (LAP and NAG) and P acquisition (AP) were calculated to document potential shifts in microbial nutrient requirements [67].

**Bacterial community composition in mesocosm soils.** The bacterial communities were characterized by barcoded amplicon sequencing using the Illumina MiSeq platform (Argonne National Laboratory, Illinois USA). We targeted the bacterial V4 region (primers 515F/806R) [68]. The complete process of library preparation and sequencing was described previously [69].

Merged paired-end reads were quality filtered (max. expected error rate 0.01, no ambiguous bases, min. length 252 bp) and trimmed to a length of 252 bp. OTUs were clustered at a 100% similarity threshold (singletons discarded) using USEARCH v11 [70]. BLAST algorithm [71] through parallel_assign_blast.py script of QIIME v 1.9.1 pipeline [72] and database Silva 138 [73] were used for taxonomy annotation. One sample with < 2000 reads was omitted from further analyses, because such a low sequencing depth might be insufficient to cover the main pattern of the community composition [68]. The raw reads were deposited in NCBI Sequence Read Archive (SRA) database under BioProject PRJNA800209 as submission SUB10992790.

## Statistics

Arithmetic means and standard deviations were calculated for plant and soil characteristics from the *in situ* controls (n = 3 for sites L and H) and mesocosm samples (n = 3 at *in situ* H and the transferred H→L) for each valley. In the case of nutrient leaching, the corrected and recalculated amounts of N and P trapped in the ion-exchange resins placed under the three H and H→L mesocosms were always averaged. Plant C:N and C:P ratios were calculated on a molar basis. The raw data of all the measured plant and soil characteristics are available in S1 Table.

The t-tests were used to evaluate (1) the "placing-to-box" effect, using the soil properties of mesocosms and ambient soils sampled in 2015, and (2) the effect of the downward transfer on plant and soil properties, using values from the H and transferred H→L mesocosms. Before the statistical analyses, the normality of each data product (i.e., plant and soil characteristic) was checked using histograms and the Kolmogorov-Smirnov test. The homogeneity of variances was verified using Bartlett's test. Data were log transformed when necessary to meet the assumptions of the t-test. Linear regressions were used to test relationships between nutrient concentrations in plant biomass and leaching from soils enclosed in the respective mesocosms. Analyses were performed in Statistica 64, version 13.0 (Dell, USA).

Alpha diversity indices (OTU richness, Shannon and Chao1) were calculated from rarefied datasets (minimal sequencing depth 8979), while the other analyses were based on non-rarefied data [74,75]. The differences in bacterial community structures in transferred and *in situ*

mesocosms were examined by the PERMANOVA analysis (9999 permutations) of Bray-Curtis dissimilarities at the OTU level using a model considering valley and downward transfer as explanatory variables (*adonis*, R package *vegan* v. 2.5–4) [76]. The homogeneity of multivariate dispersions was confirmed prior to analysis. The effect of valley and transfer on square-root transformed relative abundances of bacterial orders was tested using generalized linear models. Post-hoc comparison of mesocosm variants was performed by estimated marginal means (*emmeans* v.1.4.1) [77].

## Results

### Different soil temperature regimes at the high and low elevation sites

The H and L sites experienced different microclimate regimes. The mean annual soil temperature at a depth of 2 cm was lower at H than L sites in both valleys, with a 4°C difference between soils at FU-H and FU-L in the more wind-exposed Furkotska valley, but only with a 0.4°C difference between VS-H and VS-L in the Velka Studena valley (Table 1). The H sites had frozen soils during the winter period, with rarely occurring thawing-freezing events in spring, while the soils at L sites generally did not freeze due to the snow insulation (Fig 2). The

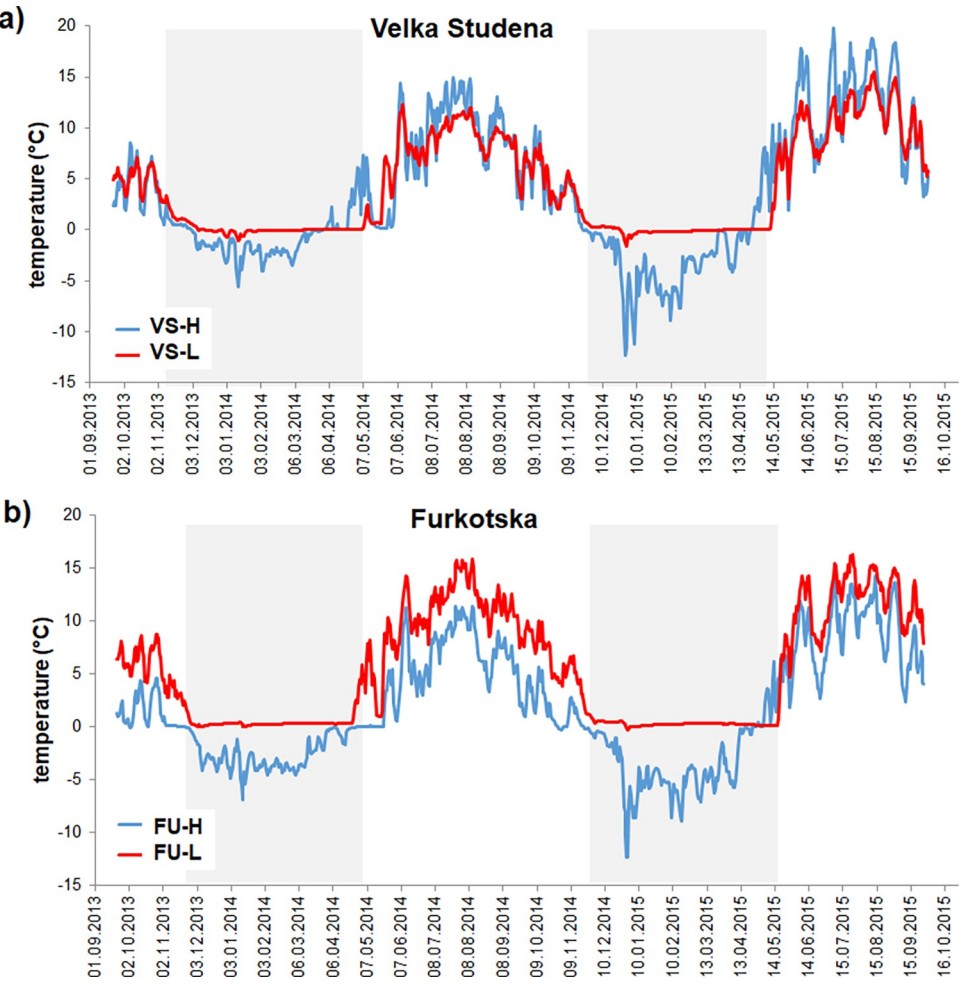

**Fig 2. Course of soil temperatures at sampling sites.** Daily mean temperature measured at the 2-cm soil depth at the high (H) and low (L) transplant sites in (a) Velka Studena (VS) and (b) Furkotska (FU) valleys during the mesocosms exposure from Sept 2013 to Sept 2015.

longer occurrence of snow cover at L sites prolonged a period of close-to-zero soil temperatures and postponed a spring soil warming by ~5–15 days, compared to the H sites, specifically in 2015 (Fig 2). During the vegetation period, the FU-H soils were cooler than those on FU-L site (Fig 2B). However, we did not observe the expected increase of temperature with decreasing elevation in the Velka Studena valley, where the sunlight exposed VS-H site had slightly warmer soils, experiencing larger diurnal temperature fluctuations than the more shaded VS-L site, located at the bottom of the deep narrow valley (Fig 2A).

## The "placing-to-box" effect

The 2-year enclosure of soils from both H sites into mesocosms did not cause significant shifts in soil properties, with few exceptions. The DOC concentration was 1.8 times higher in the ambient than enclosed soils at FU-H ($p < 0.05$) and DN concentrations (but not that of mineral N forms) were 2.7 times higher in the ambient than enclosed soils at VS-H ($p < 0.01$) (S2 Table).

## Effect of downward transfer on soil chemical properties

In both valleys, the mesocosms transfer enhanced nutrient availabilities in soil. The FU-H→L soil contained ~40% more DOC, 7-times more DN, including available mineral N forms, and ~40% more SRP compared to the FU-H mesocosms. The VS-H→L soils contained ~40% more $NH_4$-N and was 20% richer in SRP (non-significant) than the VS-H mesocosms (Table 2). Other soil physico-chemical parameters: pH, concentrations of base and acid

**Table 2. Effect of downward transfer on soil properties.** Comparison of soil physico-chemical parameters, potential activities of hydrolytic enzymes between in-site (H) and downward transferred (H→L) mesocosms after 2-year exposure in the field. Mean (standard deviations, n = 3) are shown. Characteristics, where data are given in bold differ significantly between mesocosm and ambient soils ($p < 0.05$).

| Soil property, unit | | FU-H | FU-H→L | VS-H | VS-H→L |
|---|---|---|---|---|---|
| pH | | 4.41 (0.32) | 4.65 (0.45) | 4.82 (0.18) | 4.71 (0.18) |
| CEC | meq kg$^{-1}$ | 94 (14) | 106 (10) | 123 (19) | 130 (12) |
| BS | % | 17 (4.5) | 22 (12) | 5.7 (2.1) | 6.7 (2.1) |
| $K^+$ | meq kg$^{-1}$ | 4.4 (0.82) | 4.7 (1.2) | 2.3 (0.53) | 4.3 (1.7) |
| $Na^+$ | meq kg$^{-1}$ | 0.53 (0.34) | 0.41 (0.07) | 0.45 (0.05) | 0.33 (0.06) |
| $Ca^{2+}$ | meq kg$^{-1}$ | 6.6 (2.3) | 13 (7.4) | 2.9 (1.6) | 2.7 (1.07) |
| $Mg^{2+}$ | meq kg$^{-1}$ | 3.3 (0.77) | 5.0 (1.4) | 1.3 (0.54) | 1.3 (0.21) |
| $Al^{3+}$ | meq kg$^{-1}$ | 53 (11) | 57 (16) | 90 (17) | 94 (8.8) |
| $H^+$ | meq kg$^{-1}$ | 26 (1.8) | 26 (4.5) | 26 (2.7) | 27 (3.8) |
| C | mg kg$^{-1}$ | 90 (32) | 111 (12) | 45 (19) | 34 (0.63) |
| N | mg kg$^{-1}$ | 6.7 (2.2) | 7.9 (0.7) | 3.2 (1.5) | 2.3 (0.4) |
| P | mg kg$^{-1}$ | 0.02 (0.01) | 0.01 (0.00) | 0.01 (0.00) | 0.01 (0.00) |
| DOC | mg kg$^{-1}$ | **96 (3.3)** | **135 (33)** | 42 (13) | 44 (11) |
| DN | mg kg$^{-1}$ | **6.4 (1.5)** | **44 (14)** | 3.7 (1.4) | 3.2 (0.8) |
| NH4-N | mg kg$^{-1}$ | **1.1 (0.5)** | **2.7 (1.6)** | **0.59 (0.20)** | **0.83 (0.20)** |
| NO3-N | mg kg$^{-1}$ | **2.7 (1.3)** | **5.6 (2.3)** | 2.4 (1.1) | 1.7 (0.50) |
| Nmin | mg kg$^{-1}$ | **4.1 (1.8)** | **8.3 (1.7)** | 3.0 (1.2) | 2.5 (0.5) |
| SRP | mg kg$^{-1}$ | **0.26 (0.07)** | **0.36 (0.06)** | 0.07 (0.01) | 0.09 (0.01) |
| β-glucosidase | μmol g$^{-1}$ h$^{-1}$ | 1.5 (0.31) | 1.4 (0.39) | **0.79 (0.33)** | **0.46 (0.03)** |
| Cellobiosidase | μmol g$^{-1}$ h$^{-1}$ | 0.41 (0.13) | 0.37 (0.12) | **0.17 (0.09)** | **0.08 (0.01)** |
| Phosphatase | μmol g$^{-1}$ h$^{-1}$ | 1.4 (0.50) | 1.7 (0.54) | 1.6 (0.37) | 1.2 (0.16) |
| Ala-aminopeptidase | μmol g$^{-1}$ h$^{-1}$ | 0.03 (0.01) | 0.05 (0.01) | 0.02 (0.01) | 0.02 (0.01) |
| Chitinase | μmol g$^{-1}$ h$^{-1}$ | 0.10 (0.01) | 0.13 (0.04) | 0.09 (0.01) | 0.07 (0.01) |

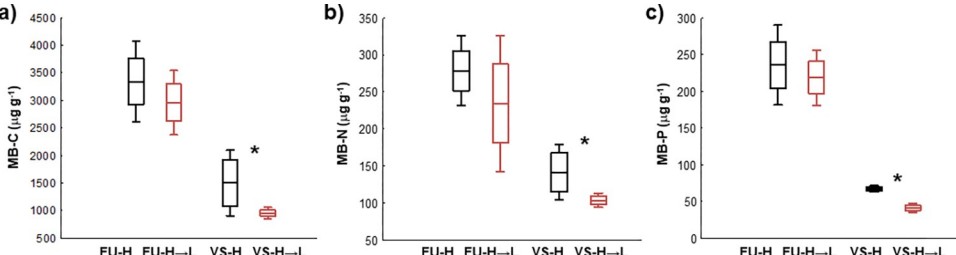

**Fig 3. Effect of downward transfer of mesocosms on soil microbial biomass (MB).** Concentrations of C (a), N (b), and P (c) in soil microbial biomass (MB) in the high (H) and transferred (H→L) mesocosms in Furkotska (FU) and Velka Studena (VS) valleys (mean as line, standard error as box and standard deviation as whiskers are shown, n = 3). Asterisks mark significant effect of downward transfer on microbial biomass ($p < 0.05$).

cations, CEC, and total C, N, and P contents were not affected by the downward transfer (Table 2).

## Effect of downward soil transfer on microbial biomass and activities

The downward transfer of mesocosms reduced the soil microbial biomass in comparison to control mesocosms (Fig 3). The soil concentrations of MB-C, MB-N and MB-P in the FU-H→L mesocosms decreased by 7–15% compared to the FU-H mesocosms, but these differences were not significant. In the VS-H→L mesocosms, the significant decreases in MB-C, MB-N, and MB-P concentrations occurred in comparison to the VS-H control mesocosms (Fig 3).

Total activity of hydrolytic enzymes decreased significantly in the VS-H→L compared to VS-H mesocosms ($p < 0.05$), while it did not change in the FU-H→L in comparison to the respective control mesocosm (Fig 4A). Activities of individual enzymes varied in response to transfer, but both C-mining enzymes (BG+CB) systematically decreased in the transferred VS-H→L (significantly) and FU-H→L (non-significantly) mesocosms (Table 2). Therefore, the relative enzymatic investment into C-mining decreased in favor of P-mining phosphatases in all downward transferred mesocosms (Fig 4B).

Potential C and N mineralization rates were accelerated and more variable in the mesocosms transferred downhill compared to the respective *in situ* H controls in both valleys (Fig

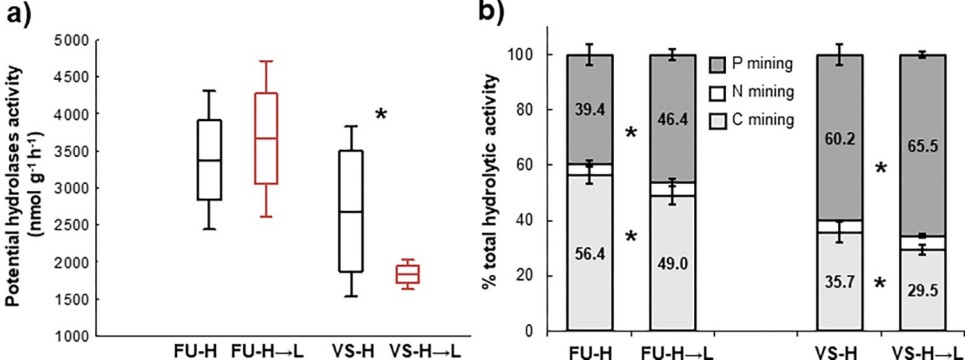

**Fig 4. Effect of downward transfer on activity of hydrolytic enzymes.** (a) Total potential activity of hydrolytic exoenzymes (mean as line, standard error as box and standard deviation as whiskers are given, n = 3) and (b) proportional investments in C, N and P mining (mean standard deviations are given, n = 3) in the soils of *in situ* H and downward transferred H→L mesocosms. Asterisks mark the significant effect of downward transfer on depicted characteristics ($p < 0.05$). Abbreviations: FU, Furkotska valley; VS, Velka Studena valley.

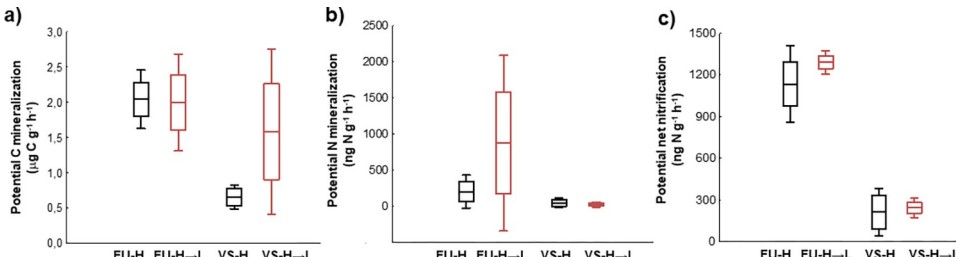

**Fig 5. Effect of downward transfer on microbial processes.** (a) Rates of potential C mineralization, (b) potential net N mineralization and (c) net nitrification in the *in situ* H and transferred H→L mesocosms (mean as line, standard error as box and standard deviation as whiskers are shown, n = 3). Abbreviations: FU, Furkotska valley; VS, Velka Studena valley.

5A and 5B). However, the increase was not significant due to the large variability in data. The transfer did not influence net nitrification rates (Fig 5C).

## Effect of downward soil transfer on bacterial community composition

The composition of soil bacterial communities differed between the control *in situ* mesocosms in the two valleys (S3 Table). Mesocosms in the FU have a higher proportion of Ktedonobacterales, Chitinophagales, Caulobacterales, and Armatimonadales. Soils in the VS were richer in Acidobacteria (Subgroup 2) and Solibacterales, Chthoniobacteriales, and RCP2-54 (S4 Table). The downhill transfer of mesocosms did not significantly affect the overall composition of bacterial communities and diversity indices (S3 Table). Therefore, we did not conduct further tests on the transfer effect on the abundance of the respective bacterial orders.

## Effect of downward soil transfer on *in situ* P and N leaching

The downward transfer systematically accelerated the P leaching from the FU-H→L mesocosms in comparison to the respective *in situ* H transplants in both years (Fig 6A).

The N leaching (the sum of $NH_4$-N and $NO_3$-N losses) composed from 60–80% of leached $NO_3$-N at all sites. Generally, the N leaching from the soils was an order of magnitude higher than the P leaching and higher in 2014 than in 2015. The transferred FU-H→L and VS-H→L mesocosms showed enhanced N leaching compared to the *in situ* H mesocosms in both years, with a significant increase only in the FU (Fig 6B).

## Nutrient concentrations in plant biomass in mesocosms

The plant biomass in the FU-H→L and VS-H→L mesocosms was enriched in N and P, having significantly lower C/P and C/N ratios than plants in the respective *in situ* H mesocosms (Fig 7A and 7B). The N and P concentration in the above- and belowground plant biomass correlated with the amounts of the nutrients leached from the respective mesocosms (Fig 7C and 7D).

## Discussion

Alpine and arctic systems are characterized by harsh climate conditions and considerable seasonality. The growing season can be up to 50% longer below- than aboveground due to the temperature-buffering capacity of the soils in such systems [29]. Therefore, any change in soil temperature regime would influence the timing of growth and activity of root and associated microorganisms, and rates of biogeochemical processes, which may result in significant changes in the functioning of the entire system.

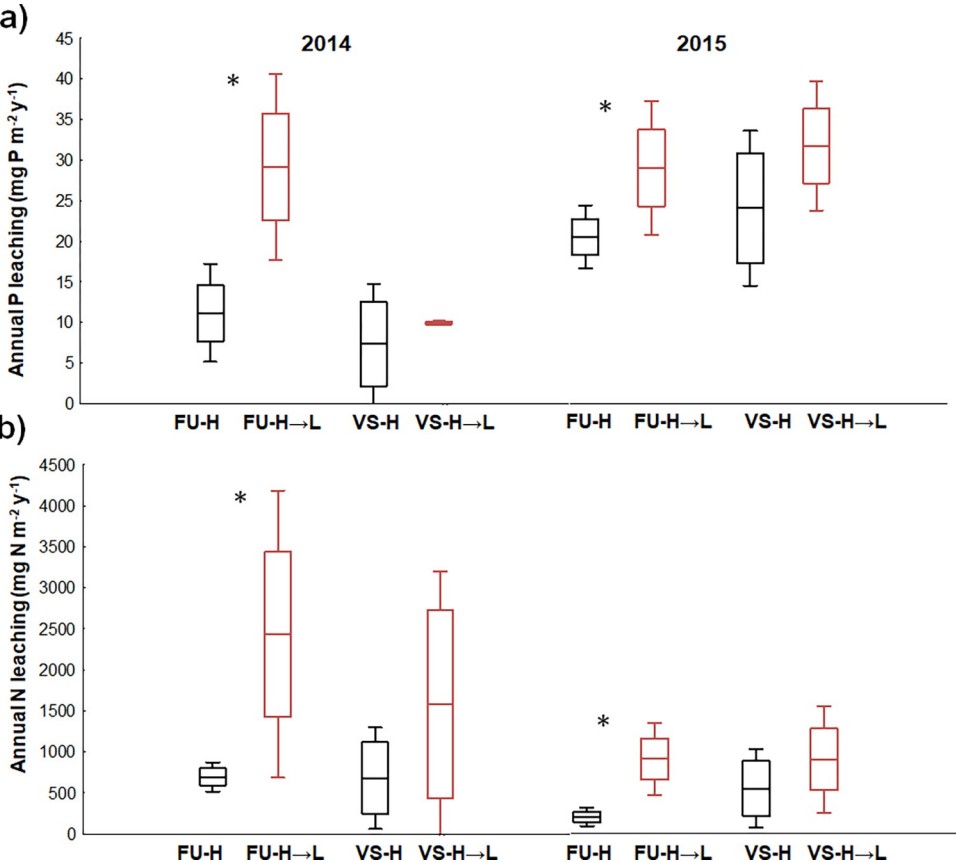

**Fig 6. Effect of downward transfer on nutrient leaching.** *In situ* (a) P and (b) N leaching from original and downward transferred soils in Furkotska (FU) and Velka Studena (VS) valleys in 2014 (Sept 2013–Sept 2014) and 2015 (Sept 2014–Sept 2015). N leaching composes of ammonium- and nitrate-N leaching (means and standard deviations are given, n = 3). Asterisks mark significant changes in element leaching between the *in situ* H and transferred H→L mesocosms in particular year ($p < 0.05$).

Through the downward transfer of plant-soil mesocosms, we simulated the climate shift from harsh conditions prevailing in the alpine Tatra Mountain zone at ~2100 m a.s.l. into the milder temperature regime in the subalpine zone. In addition, the highly wind-exposed alpine meadows at the H sites commonly froze during the winter due to lower temperature and thinner snow cover. Snow blown from the steep, higher elevation meadows accumulated in more wind-protected areas where the L sites were located. Therefore, the downward transferred mesocosms experienced steadily higher soil temperature than the H sites during the whole study, including the winter period with daily average temperatures $\geq 0°C$ (Fig 2). The downward transferred mesocosms in the FU (although not in the VS valley) was additionally exposed to significant soil warming during the vegetation period (Fig 2). Despite the relatively short period of the experiment, the mesocosms' 2-year exposure to the warmer temperature regime significantly altered their soil nutrient cycling. All the H→L mesocosms showed evidence of fastened nutrient cycling and elevated N and P leaching in comparison to their respective H controls.

Plant biomass in the transferred mesocosms was enriched with nutrients, namely in terms of the P content (Fig 7B). Plants take up and immobilize nutrients in their biomass during the whole vegetation season and are also capable of winter N and P uptake [78–80], specifically in alpine and arctic regions, where roots remain active longer than shoots due to the snow's

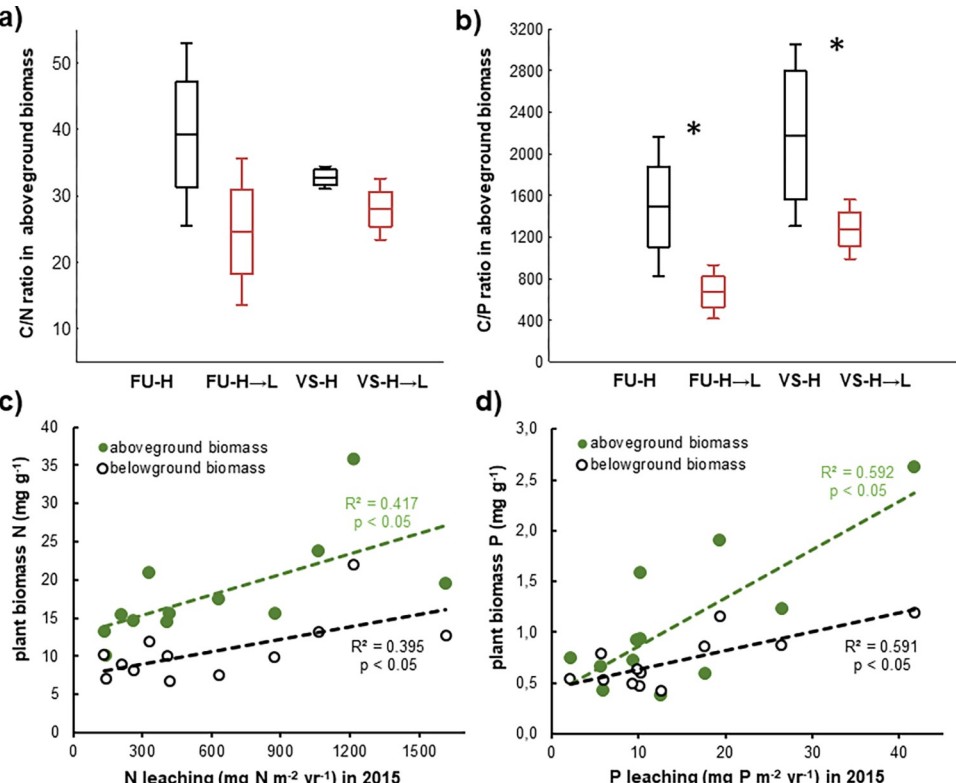

**Fig 7.** The plant tissue C/N (a) and C/P (b) ratios in the *in situ* H and transferred H→L mesocosms in Furkotska (FU) and Velka Studena (VS) valleys in 2015 (mean as line, standard error as box and standard deviation as whiskers are given, n = 3), asterisks mark significant effect of downward transfer ($p < 0.05$). Correlations between N (c) and P (d) concentration in the above- and belowground plant biomass with the amount of mineral N and P, respectively, contained in ion-exchange resin traps in 2015.

insulating capacity [29]. The nutrient-enriched plant biomass in the H→L mesocosms, which integrated the whole-year nutritional state of the soils, thus served as a useful and sensitive indicator of the enhanced *in situ* soil N and P availability.

In agreement with the plant indications of enhanced *in situ* nutrient availability, the soils of all H→L mesocosms contained more SRP and available $NH_4$-N than their H controls (Table 2). The soils of FU-H→L mesocosms, facing higher temperatures also during vegetation periods (Fig 2A), were additionally enriched in total available N (DN, $NO_3$-N and $NH_4$-N), and DOC in comparison to the H controls (Table 2) at the end of the 2015 vegetation period. Specifically the increases in DOC and available N concentrations in the soils from FU-H→L mesocosms are likely related to the enhanced plant activity, which is commonly observed during warmer vegetation periods [81–83]. The higher plant activity is directly responsible for higher input of organic compounds to the soil [84], which contributes to the DOC pool [85], stimulates soil microbial activity, and enhances N recycling and plant N supply [86].

The observed higher nutrient availability in the systems exposed to higher temperature was in agreement with results of other ecosystem warming experiments performed elsewhere [16,18], as well as with snow manipulation and transplant studies [87–89]. The larger pools of available N in "warmer" soils are commonly associated with higher rates of gross and net N mineralization, respiration and enzymatic activity [16,37,87,89,90]. However, in contrast to those studies, we did not observe significant alteration of soil processes, like rates of total

exoenzymatic activity and C and N mineralization and nitrification, in the "warmer" H→L transplants compared to the *in situ* H controls, when measured in the standardized laboratory conditions. The absence of the functional adaptation after the two-year exposure to a milder climate regime was in accord with the stable composition of the soil microbial community. Similar structural continuity of the soil microbial communities were observed also in other studies of "climate change in cold ecosystems": e.g. in the four-year transplant study conducted in subalpine grassland [30], and even in 17-year long transplant experiment in mountain grassland [19]. Similarly, Rinnan et al. [91] showed that >10-year period was needed for the development of significant changes in microbial biomass and strong alterations in microbial community composition in the 15-year lasting climate change manipulations in a subarctic heath ecosystem. They also showed that bacterial communities from colder regions were less temperature sensitive than those from the warmer regions by comparing bacterial growth rates and turnover in terrestrial Antarctic ecosystems [92].

The fact that we did not observe any structural and functional adaptation of the soil microbial community after the 2-year experiment suggests that warmer winter periods (Fig 2) were critical in enhancing nutrient availability in the downward transplants. While the constantly below-zero soil temperatures limited the overall microbial activity and nutrient mineralization at the H sites (Fig 2), the close-to-zero temperatures in the more snow-insulated soils in the H→L mesocosms allowed microorganisms to remain active during the whole winter, as also observed elsewhere [88,93,94]. Such soil warming during winter period, connected with efficient snow insulation [37,40,41], was shown to enhance N availability in the system in spring. In our case, the "extra" released N in the H→L mesocosms, which remained active for 4–6 winter months longer than the H mesocosms, was available to the plants at the beginning of the growing season and was partly immobilized in their biomass (Fig 7A). The rest of potentially available N was washed out of the system and retained in ion-exchange resin traps under the mesocosms (Fig 6B). Elevated N losses from the systems exposed to climate warming were reported in several other studies, e.g., from temperate heathlands and grasslands [89,90], oceanic mountain ecosystems [41] or forests [43]. Some of the studies [41,43] additionally show that the N leaching rates from the warmed systems are high especially at sites with persistent winter snow cover in comparison to not-insulated sites exposed to frequent soil freeze-thaw events.

In contrast to N cycling, the knowledge of warming effects on the soil P transformation and availability is limited. Our study provides the first *in situ* evidence showing that the increasing average annual temperature enhances the P availability in soils (Table 2), resulting in P enrichment in plant biomass (Fig 7B) and larger P leaching (Fig 6A). It is likely that winter microbial activity contributes to the accumulation of available P in the soil, some of which is flushed out with water from thawing snow, similarly to mineral N. Consistent with the increased soil P availability in the H→L mesocosms, we observed a systematic decrease in the soil enzymatic C:P ratio in the late growing season (Fig 4B). It resulted from a reduced enzymatic C acquisition and increased activity of phosphatases. Such change in C:P acquisition could be associated with greater plant productivity and organic C exudation due to enhanced nutrient supply (Fig 7, Table 2) and higher temperatures during vegetation period (in case of the FU), as observed also elsewhere [82].

The mineral N and P leaching (quantified by ion-exchange resin traps) correlated with the N and P concentrations in plant biomass (Fig 7C and 7D). Both these methods integrate the *in situ* nutrient availability in the whole plant-soil system for long-term periods (vegetation season for plant biomass and annual element leaching for ion-exchange resin traps). This is their major advantage over seasonally changing parameters such as soil chemistry, microbial activity, and active microbial community composition that could confound the response of plant-

soil systems under changing climate conditions. Both methods, ion-exchange resin traps quantifying nutrient leaching and nutrient enrichment of plant biomass indicating enhanced N and P availability, have been demonstrated to be sensitive indicators of the accelerated soil N and P cycling in the H→L mesocosms even in the absence of any supportive significant changes in microbial biomass, the structure of bacterial community, or C and N mineralization rates. The elevated losses of mineral N and P forms from the alpine grasslands documented in our transplant experiment complement earlier observations in the alpine catchments in the Tatra Mountains [52,53], showing the complex effect of climate change on soil biogeochemical processes. We provide the evidence that the N and P cycles accelerate in alpine meadow soils, most likely due to continuing microbial activity in snow-insulated soils during winter and increased plant activity in the growing season.

Ongoing climate change will lead to rising temperatures in the growing season and also in winter, combined with a decrease in snow precipitation and accumulation. Overall warmer winters and thinner and discontinuous snowpack will likely result in reduced snow insulation, more frequent freeze-thaw events in soils, and their washing with infiltrating water. These conditions will result in repeating cycles of increased microbial activity and decay and leaching of mineral nutrient forms already during winter. The final amount of nutrients accumulated in the soil and available in the beginning of the growing season will thus increasingly depend on the winter weather condition [41,43], especially at elevations with winter air temperature fluctuating around freezing point. It will inevitably lead to an increase in year-on-year variability in soil nutrient availability and leaching to surface waters depending on current weather conditions.

## Conclusions

The use of a transplant method allowed us to study the complex response of the alpine system to changing climate conditions, with both the plants and the soil microbiome responding inseparably and in close interaction.

Alpine meadow ecosystems sensitively responded to climate change simulated by a downward transfer along the elevation gradient. Even in the short term, the increase in the mean annual air temperature by 2˚C accelerated the nutrient cycling, shown by a higher content of N and P in plant biomass and greater annual losses of these nutrients from the systems by leaching. Except signs of enhanced plant root activities and preferential enzymatic mining of P versus C, changes in other soil characteristics such as bacterial community composition, microbial biomass and potential rates of decomposition and nutrient transformation processes were not significant. These results suggest that prolonged, warmer winter periods due to snow insulation, which enabled soil microorganisms to remain active, were critical for enhanced nutrient availability in the alpine meadows.

The duration of the experiment was not suitable for tracking longer-term changes in the ecosystem functions associated with potential shifts in plant vegetation composition, litter quality, and microbial adaptation to the new conditions. However, we can expect that ongoing climate change will lead to a decrease in snowfall, depth and duration of snow cover, resulting in more frequent freeze-thaw events in soils that have so far been frozen or covered with snow. This will affect the nutrient cycling in alpine meadow systems compared to the current situation, and the year-on-year variability of soil nutrient availability and leaching from alpine and subalpine meadows will probably increase.

## Supporting information

**S1 Table. Raw data characterizing soil and plant chemistry a soil biochemistry in transplants left in site and mowed downward in two valleys in the Tatra Mountains.** Data were

obtained during the destructive sampling in September 2015, 2 years after exposure of the transplants to the field conditions. The data were used to calculate means and standard deviations given in the paper.
(DOCX)

**S2 Table. Comparison of physico-chemical and biochemical characteristics of mesocosm and ambient soils sampled at the end of the 2-year experiment in September 2015.** Mean (standard deviations, n = 3) are shown. Characteristics where data are given in bold differ significantly between mesocosm and free soils ($p < 0.05$).
(DOCX)

**S3 Table. Effect of valley, downward transfer and their interaction on mesocosm soil bacterial community composition–results of PERMANOVA.**
(DOCX)

**S4 Table. The most abundant bacterial orders ($> 1\%$ in at least one variant) and alpha diversity measures in *in situ* and transferred mesocosms.** Numbers represent mean (relative abundance in % in case of orders) with standard deviation in brackets. $\chi2$ and p-value for test of differences between valleys are presented. Small letters denote significant differences at $p < 0.05$.
(DOCX)

## Acknowledgments

We thank Ondra Žampach, Dan Vaněk, Katka Kučerová and Hana Petrásková who assisted us in laboratory analyses and the authorities of the Tatra National Park (TANAP) in Slovakia for their administrative support. We further thank Ryan Scott and Gabriela Scott Zemanová for the proofreading.

## Author Contributions

**Conceptualization:** Eva Kaštovská, Jiří Kopáček.

**Data curation:** Michal Choma, Petr Čapek, Jiří Kaňa, Karolina Tahovská.

**Formal analysis:** Michal Choma, Petr Čapek, Karolina Tahovská.

**Funding acquisition:** Jiří Kaňa.

**Investigation:** Michal Choma, Petr Čapek, Jiří Kaňa.

**Methodology:** Eva Kaštovská, Jiří Kaňa, Jiří Kopáček.

**Project administration:** Jiří Kaňa.

**Supervision:** Jiří Kopáček.

**Writing – original draft:** Eva Kaštovská.

**Writing – review & editing:** Eva Kaštovská, Michal Choma, Petr Čapek, Jiří Kaňa, Karolina Tahovská, Jiří Kopáček.

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
