## [Decision Letter · Decision Letter 0]

31 May 2022

PONE-D-22-06781Soil warming during winter period enhanced soil N and P availability and leaching in alpine grasslands: a transplant studyPLOS ONE

Dear Dr. Kastovska,

Thank you for submitting your manuscript to PLOS ONE. After careful consideration, we feel that it has merit but does not fully meet PLOS ONE’s publication criteria as it currently stands. Therefore, we invite you to submit a revised version of the manuscript that addresses the points raised during the review process. Please submit your revised manuscript by Jul 15 2022 11:59PM. If you will need more time than this to complete your revisions, please reply to this message or contact the journal office at plosone@plos.org. Please include the following items when submitting your revised manuscript:A rebuttal letter that responds to each point raised by the academic editor and reviewer(s). You should upload this letter as a separate file labeled 'Response to Reviewers'.A marked-up copy of your manuscript that highlights changes made to the original version. You should upload this as a separate file labeled 'Revised Manuscript with Track Changes'.An unmarked version of your revised paper without tracked changes. You should upload this as a separate file labeled 'Manuscript'.If applicable, we recommend that you deposit your laboratory protocols in protocols.io to enhance the reproducibility of your results. Protocols.io assigns your protocol its own identifier (DOI) so that it can be cited independently in the future. For instructions see: https://journals.plos.org/plosone/s/submission-guidelines#loc-laboratory-protocols. Additionally, PLOS ONE offers an option for publishing peer-reviewed Lab Protocol articles, which describe protocols hosted on protocols.io. Read more information on sharing protocols at https://plos.org/protocols?utm_medium=editorial-email&utm_source=authorletters&utm_campaign=protocols.

We look forward to receiving your revised manuscript.

Kind regards,

Iván Prieto Aguilar, Ph.D.

Academic Editor

PLOS ONE

Journal Requirements:

2. In your Methods section, please provide additional information regarding the permits you obtained to collect samples for the present study. Please ensure you have included the full name of the authority that approved the field site access and, if no permits were required, a brief statement explaining why.

Additional Editor Comments:

Dear Eva Kastovska,

This manuscript deals with an interesting topic evaluating soil microbial responses to changing climatic conditions. The manuscript is sound but it is important to clarify any potentially confounding effects of soil transplantations to lower altitudes and I also find it interesting, given that the data collected is suitable for such analyses, to assess potential changes in soil microbial diversity and functionallity with increasing temperatures.

Best regards,

Reviewers' comments:

Reviewer's Responses to Questions

**Comments to the Author**

1. Is the manuscript technically sound, and do the data support the conclusions?

Reviewer #1: Partly

Reviewer #2: Yes

2. Has the statistical analysis been performed appropriately and rigorously? 

Reviewer #1: Yes

Reviewer #2: Yes

3. Have the authors made all data underlying the findings in their manuscript fully available?

Reviewer #1: Yes

Reviewer #2: Yes

4. Is the manuscript presented in an intelligible fashion and written in standard English?

Reviewer #1: Yes

Reviewer #2: Yes

5. Review Comments to the Author

Reviewer #1: This manuscript evaluates the effects of warming in soil microbial communities and functional processes. For this purpose, authors transplanted soils from high mountain points to lower positions with warmer temperature. The experimental design is well executed. Introduction is well elaborated and supports very interesting hypotheses.

Major comments to be acknowledge in the manuscript:

i) Can authors disentangle the effects of warming from other related effects (i.e. plant species, litter, etc.) when transplanting soil?

ii) Only one sampling time is considered. Is only one sampling time enough to conclude about the effects of warming? Or can seasonal effects confound these responses?

Minor suggestions:

L26: be more specific with “this changes”.

L34: “active” in which terms?

L42: maybe add “with associated changes in microbial biomass”

L105-111: consider move to M&M

Reviewer #2: The paper by Kaštovská et al. titled: “Soil warming during winter period enhanced soil N and P availability and leaching in alpine grasslands: a transplant study” is an interesting study on how Alpine meadows are strongly affected by climate change.

To estimate the effect of this change on soil nutrient cycling, the authors conducted a field experiment where they transferred undisturbed plant-soil mesocosms from two wind-exposed alpine meadows at ~2100 m a.s.l. to more sheltered plots, situated ~300–400 m lower in the same valleys.

The paper is generally very well written, the material and methods are very well detailed also with a correct and comprehensive statistical analysis. The results are very well described and well discussed. I have enjoyed reading this manuscript. I have not many suggestions to make to improve this work. Just few comments:

-Move the caption figures (Fig. 1 to 7) at the end of the manuscript after references

-Move the tables also at the end of the manuscript before of caption figures.

-In line 199 and 337 change “basic soil properties” by chemical soil properties.

-The authors found that the downhill transfer of mesocosms did not affect the bacterial community composition; however I am surprised since the bacteria are very sensitive to changes. Sometimes although we don´t observe changes in the bacterial community composition there are changes in the diversity index (Shannon, pielou, richness etc) also in the relative abundance of any bacterial phylum or family but I don’t see anywhere that the authors had done these analysis. I suggest performing this analysis diversity to check. They will improve the manuscript quality if the authors find any significant difference in their study.

On the other hand, the functional diversity has recently begun to be examined to understand the functional aspects of microbial ecology. The functional potential of a bacterial community can be inferred through direct observation of functional genes and pathways by whole metagenome sequencing surveys or indirectly by marker-gene surveys. To do this, several computational approaches have been developed which infer the approximate functional potential from taxonomic profiles obtained from amplicon sequencing (i.e., functions encoded at the DNA level). For example, PICRUSt2 (Phylogenetic Investigation of Communities by Reconstruction of Unobserved States) performs marker gene metagenome inference based on the 16S rRNA gene.

If the authors find any change in the diversity analysis suggested before I encourage the authors to carry out this bacterial functional analysis as well.

6. PLOS authors have the option to publish the peer review history of their article (what does this mean?). If published, this will include your full peer review and any attached files.

Reviewer #1: No

Reviewer #2: No

---

## [Author Response · Author response to Decision Letter 0]

7 Jul 2022

Please, find our comments to reviewers and editor in a rebuttal letter, which is a part of submittion.

Thank you.

---

## [Editor Report · Decision Letter 1]

14 Jul 2022

Soil warming during winter period enhanced soil N and P availability and leaching in alpine grasslands: a transplant study

PONE-D-22-06781R1

Dear Dr. Kastovska,

We’re pleased to inform you that your manuscript has been judged scientifically suitable for publication and will be formally accepted for publication once it meets all outstanding technical requirements.

Kind regards,

Iván Prieto Aguilar, Ph.D.

Academic Editor

PLOS ONE

Additional Editor Comments (optional):

Thank you for your contribution, it has resulted in a very nice paper evaluating plant responses to climate change.
---

## [Editor Report · Acceptance letter]

25 Jul 2022

PONE-D-22-06781R1 

Soil warming during winter period enhanced soil N and P availability and leaching in alpine grasslands: a transplant study 

Dear Dr. Kaštovská:

I'm pleased to inform you that your manuscript has been deemed suitable for publication in PLOS ONE. Congratulations! Your manuscript is now with our production department. 

Kind regards, 

on behalf of

Dr. Iván Prieto Aguilar 

Academic Editor

PLOS ONE